# Prophylactic Central Lymph Node Dissection Improves Disease-Free Survival in Patients with Intermediate and High Risk Differentiated Thyroid Carcinoma: A Retrospective Analysis on 399 Patients

**DOI:** 10.3390/cancers12061658

**Published:** 2020-06-23

**Authors:** Fabio Medas, Gian Luigi Canu, Federico Cappellacci, Giacomo Anedda, Giovanni Conzo, Enrico Erdas, Pietro Giorgio Calò

**Affiliations:** 1Department of Surgical Sciences, University of Cagliari, 09124 Cagliari, Italy; gianlu_5@hotmail.it (G.L.C.); fedcapp94@gmail.com (F.C.); ganeddamed@gmail.com (G.A.); erdasenrico@libero.it (E.E.); pgcalo@unica.it (P.G.C.); 2Division of General and Oncologic Surgery, Department of Cardiothoracic Sciences, University of Campania “Luigi Vanvitelli”, 80131 Naples, Italy; giovanni.conzo@unicampania.it

**Keywords:** thyroid carcinoma, prophylactic central lymph node dissection, lymph node metastases

## Abstract

The role of prophylactic central lymph node dissection (pCLND) in the treatment of differentiated thyroid cancer (DTC) is controversial and still a matter of debate. The primary outcome of our study was to assess whether pCLND is effective in reducing the incidence of recurrent disease, and the secondary goal was to estimate the incidence of postoperative complications in patients who underwent pCLND and to evaluate the prognostic value of occult node metastases. In this retrospective study, we included patients with preoperative diagnosis of DTC and clinically uninvolved lymph nodes (cN0). The patients were divided into two groups, depending on the surgical approach: total thyroidectomy alone (TT group) or total thyroidectomy and pCLND (pCLND group). Three hundred and ninety-nine patients were included in this study, 320 (80.2%) in the TT group and 79 (19.8%) in the pCLND group. There were no significant differences in morbidity among the two groups. Histopathological evaluation demonstrated a similar distribution of aggressive features, especially regarding multicentricity, extrathyroidal extension, and angioinvasivity between the two groups. Occult lymph node metastases were found in 20 (25.3%) patients in the pCLND group. Prophylactic CLND was effective in improving disease-free survival in patients with intermediate and high risk of disease recurrence (*p* = 0.0392); occult lymph node metastases resulted as a significant negative prognostic factor (*p* < 0.001).

## 1. Introduction

Differentiated thyroid carcinoma (DTC) is traditionally considered as a tumor with good prognosis, with an overall survival nearly comparable to the general population. Nevertheless, a certain number of patients experience a poor clinical outcome, with local recurrence requiring further medical or surgical treatment, with a considerable worsening of the quality of life. 

The incidence of local recurrence is widely variable; the American Thyroid Association (ATA) guidelines for patients with differentiated thyroid cancers, published in 2015, report an incidence of local recurrence in 3–13% of patients with low-risk tumors, 21–36% in cases of intermediate-risk tumors, and in 68% of high risk tumors [1]; similar findings have been extensively reported in literature [2,3,4,5,6,7].

The most important risk factors for local recurrence have been reported in the literature and are the presence of lymph node metastases, the extrathyroidal extension of the tumor with invasion of perithyroidal tissues, the presence of BRAF V600E mutation, and the incomplete resection of the tumor [1,2,8,9].

In fact, one of the outstanding issues is the role of lymphectomy of the central compartment in cases of clinically uninvolved lymph nodes. In fact, the incidence of occult node metastases is high, with a reported incidence up to 90% [2,10,11,12,13,14]. Furthermore, the real prognostic value of lymph node metastases in DTC is still uncertain, with some authors suggesting that lymph node metastases do not decrease the survival rate [3,15,16,17,18], while others have reported a worsening in both overall and in disease-free survival [8,19,20,21,22,23].

For these reasons, a prophylactic central lymph node dissection (pCLND) has been proposed for tumors with clinically uninvolved lymph nodes, but suspected of having aggressive behavior at preoperative and intraoperative evaluation. Nevertheless, this suggestion is met with considerable resistance due to the higher incidence of postoperative complications including hypoparathyroidism and recurrent laryngeal nerve (RLN) injury, and to the doubtful role of lymph node metastases on prognosis [24,25,26]. Consequently, the most important guidelines on DTC are discordant on this topic, with the ATA guidelines and National Comprehensive Cancer Network (NCCN) guidelines suggesting a prudent approach [1,27], while the Japanese Association of Endocrine Surgeon and the Japanese Society of Thyroid Surgeons recommend routine pCLND [28].

The aim of this study was to assess whether prophylactic CLND is effective in reducing the incidence of recurrent disease, to evaluate the incidence of postoperative complications in patients who underwent pCLND, and the influence of occult node metastases on the prognosis.

## 2. Methods

In this retrospective observational study, we included patients who underwent thyroidectomy for DTC at our department from January 2011 to December 2016. Ethical approval was obtained from our local ethics committee. Patients were identified from a prospectively maintained institutional database including all patients who underwent thyroidectomy. Inclusion criteria were preoperative diagnosis of DTC and the absence of clinically involved lymph nodes both on Ultrasound (US) and on intraoperative examination (cN0). Exclusion criteria were distant metastases, incidental diagnosis of DTC at pathological evaluation, and surgery performed for recurrent disease.

Patients were divided into two groups: in the first group, we included patients who underwent total thyroidectomy alone (TT group), while in the second group, patients who had undergone thyroidectomy and prophylactic CLND (pCLND group). The design of the study is reported in Figure 1.

### 2.1. Outcome of the Study

The primary outcome of the study was to evaluate whether prophylactic CLND improves disease-free survival in patients with DTC. The secondary outcomes were to assess whether prophylactic CLND is burdened by a higher incidence of postoperative complications, specifically RLN injury and hypoparathyroidism, and to evaluate the prognostic significance of occult lymph node metastases.

### 2.2. Preoperative Evaluation

Preoperative evaluation included clinical history, physical examination, and blood tests to assess thyroid function and autoimmune thyroiditis. Fine-needle aspiration cytology (FNAC) was performed in all patients, and the results were classified according to the Consensus Statement of AIT (Italian Thyroid Association), AME (Medical Endicronologist Associaction), SIE (Italian Endocrinology Association), and SIAPEC-IAP (Italian Society of Pathological Anatomy) for the Classification and Reporting of Thyroid Cytology [25]. Hyperthyroidism status was defined in the case of low serum Thyroid-Stimulating Hormone (TSH) (<0.4 mIU/L), use of thyrostatic drugs or positivity for Anti-TSH receptor antibodies (TRAb). Autoimmune thyroiditis was defined in the case of positivity for anti-thyroglobulin antibodies (Tg-Ab) or anti-thyroid peroxidase antibodies (TPO-Ab) and on the basis of histopathological examination.

High-resolution US of the neck was always performed before surgery by an experienced surgeon, with careful evaluation of the central and the lateral compartment. Preoperative laryngoscopy was routinely performed to assess vocal fold mobility.

Indication for surgery was preoperative cytologic diagnosis or suspicion of DTC or, in the case of negative cytology, the presence of a highly suspicious nodule based on family history, physical examination, and US features of the nodule.

### 2.3. Surgical Treatment

All procedures were performed by three endocrine surgeons with high experience in thyroid surgery, each performing at least 100 thyroidectomies per year. The surgical procedures were extracapsular total thyroidectomies. The RLNs were routinely exposed until their insertion into the larynx to avoid injury. Intraoperative neuromonitoring (IONM) of RLNs was routinely used in order to facilitate nerve identification and confirm its functional integrity. Parathyroid glands were searched at the usual sites and any attempt to preserve them was made.

Prophylactic CLND (pCLND) was performed in cases of clinically uninvolved lymph nodes (cN0) in tumors considered at high risk for recurrence based on family history, US features of the nodule, results of FNAC, and intraoperative examination of the thyroid gland, especially in cases of suspected extracapsular extension of the tumor. The choice to perform a pCNLD was planned and concerted among surgeons and endocrinologist preoperatively, and then discussed with the patient. CLND consisted of excision of all lymphatic structures included in level VI and level VII, on the basis of the recognized anatomic continuity between the superior mediastinum and neck. The anatomical limits of the dissection were represented anteriorly by the superficial layer of the deep cervical fascia, superiorly by the hyoid bone, laterally by the carotid arteries, inferiorly by the innominate artery, and posteriorly by the pre-vertebral layer of the deep cervical fascia. The central compartment included the pretracheal, prelaryngeal (Delphian), paratracheal, and paralaryngeal lymph nodes. Level VII was comprised of the superior anterior mediastinal lymph nodes, located above the innominate artery and below the level of the upper border of the sternal manubrium.

Patients in which only some perithyroidal lymph nodes were excised, without the clear intention to perform a pCLND, were included in the TT group.

### 2.4. Pathologic Examination

The surgical specimen was fixed with formaldehyde; sections were stained with hematoxylin-eosin (H&E) and analyzed by a dedicated pathologist. Immunohistochemical analysis was performed with the streptavidin-biotin technique on paraffin sections using anti-pan-cytokeratin antibody.

In cases of multifocal carcinoma, the tumor size was defined as the largest diameter among the malignant nodules. A microcarcinoma was defined as a tumor with larger diameter equal or inferior to 10 mm. Extrathyroidal extension was defined as the presence of gross infiltration of perithyroidal tissues found in pathological examination. Vascular invasion was defined as invasion of vessels in the tumor capsule or beyond it, with intravascular tumor cells attached to the vessel wall.

The lymph node yield was defined as the number of lymph nodes harvested after lymphectomy, and lymph node ratio was defined as the ratio between metastatic lymph nodes and total lymph nodes retrieved, calculated only in patients with metastatic lymph nodes.

Lymph node micrometastasis was defined as a metastasis with maximum dimension ≤0.5 cm.

### 2.5. Postoperative Management and Follow-Up

Serum calcium and (Parathyroid Hormone) PTH levels were assayed pre- and postoperatively. Postsurgical hypoparathyroidism was defined as PTH < 10 pg/mL following the operation (normal range = 10–65 pg/mL).

Postoperative fibrolaryngoscopy was performed in the case of loss of signal at IONM or in patients experiencing dysphonia after surgery, even in the case of normal signal at IONM.

Hypoparathyroidism and RLN injury were considered permanent if lasting for more than 12 months after surgery.

All patients were referred to an endocrinologist for postoperative management and were stratified in ATA groups for risk of disease recurrence, following the 2009 and then 2015 ATA Guidelines. Radioactive iodine was routinely administrated after total thyroidectomy in case of ATA intermediate and high risk tumors.

Serum thyroglobulin (Tg) and anti-Tg antibodies measurements and neck ultrasound (US) were used for postoperative evaluation. During initial follow-up, serum Tg and anti-Tg antibodies were measured every 6–12 months. More frequent Tg and anti-Tg antibody measurements were performed in ATA high risk patients. In ATA low and intermediate-risk patients that achieved an excellent response to treatment, Tg measurements were repeated every 12–24 months. ATA high risk patients (regardless of response to therapy) and all patients with biochemical incomplete, structural incomplete, or indeterminate response to treatment continued to execute Tg and anti-Tg antibodies measurements at least every 6–12 months for several years.

In patients with DTC of any risk level with significant comorbidity that precluded thyroid hormone withdrawal prior to RAI therapy, recombinant human TSH (rhTSH) preparation was done; these situations included medical or psychiatric conditions that could be acutely exacerbated in the case of hypothyroidism (leading to a serious adverse event) or inability to mount an adequate endogenous TSH response with thyroid hormone withdrawal.

Disease-free status was defined as a no evidence of disease (NED) and included the following features: no clinical evidence of tumor, no imaging evidence of disease by RAI imaging and/or neck US, and low serum Tg levels during TSH suppression (Tg < 0.2 ng/mL) or after stimulation (Tg < 1 ng/mL) in the absence of interfering antibodies.

Disease-free survival was defined as the time elapsed from surgery to the detection of recurrent disease.

For the purpose of this work, risk stratification for disease recurrence was reviewed for every patient that underwent surgery until 2015 and was eventually adapted to the latest ATA guidelines.

### 2.6. Statistical Analysis

Statistical analysis was performed with MedCalc^®^ vers. 19.2.1. The Chi-squared test and Student’s *t*-test were used for categorical and continuous variables, respectively. Log-rank test was used to estimate the differences in Kaplan–Meier curves for independent risk factors. Results were considered statistically significant in the case of a *p*-value <0.05. Continuous variables are expressed as mean ± standard deviation of the mean.

## 3. Results

We included in this study 399 patients with clinically uninvolved lymph nodes (cN0) and pathological diagnosis of DTC (Table 1). There were 101 (25.3%) males and 298 (74.7%) females with a mean age of 50.5 years. Autoimmune thyroiditis was present in 146 (36.6%) cases, and hyperthyroidism in 28 (7%). A familiar history of thyroid carcinoma was present in seven (1.8%) patients, none of these were in the context of Multiple Endocrine Neoplasia Syndrome. In 148 (37.1%) patients, FNAC demonstrated atypia of undetermined significance (including Tir3, Tir3a, and Tir3b), and in 178 (44.6%) FNAC indicated a suspicious malignancy (Tir4) or a malignant nodule (Tir5). In the remaining 73 (18.3%) patients, despite the fact that FNAC was not suspicious or indicative for malignancy, a thyroidectomy was planned based on family history of the patient, physical examination, or US features of the nodule.

The surgical procedure consisted of total thyroidectomy alone in 320 (80.2%) patients, and total thyroidectomy and prophylactic CLND in 79 (19.8%).

Patients were divided into two groups based on the surgical approach. Those who underwent prophylactic CLND were significantly younger (42.6 y.o. vs. 52.4 y.o.; *p* < 0.001) and more frequently affected with autoimmune thyroiditis (62% vs. 30.3%; *p* < 0.001). Furthermore, FNAC was significantly more frequently diagnostic for DTC in the pCLND group (69.6% vs. 38.4%; *p* < 0.001).

The mean operative time was significantly longer in the pCLND group (102.3 minutes vs. 92.5 minutes; *p* < 0.001) as well as postoperative stay (3.0 days vs. 2.8 days). The incidence of transient hypoparathyroidism (43% vs. 31.9%), permanent hypoparathyroidism (15.2% vs. 8.4%), transient RLN injury (3.8% vs. 2.2%), and permanent RLN injury (1.3% vs. 0.6%) was higher in the pCLND group, but these differences were not statistically significant. The mean follow-up time was 55.4 months.

Full histopathologic findings are reported in Table 2. Nodule size and thyroid weight were similar between the two groups. Conversely, the histotype was significantly different between the two groups (*p* < 0.001). In the pCLND group, the incidence of the tall cell variant of PTC (TCV-PTC) was nearly quadruple that in the other group (25.3% vs. 6.3%), whereas the incidence of follicular variant of PTC (FV-PTC) was less than half (16.5% vs. 37.5%) when compared to the TT group. The presence of aggressive features of the tumor including multicentricity, angioinvasivity, and extrathyroidal extension was similar between the two groups.

As defined in the Methods section, all patients in the pCLND group underwent lymphectomy of the level VI and level VII lymph nodes, whereas in the other group, an excision of some perithyroidal lymph nodes was performed in 99 (30.9%) patients.

Lymph node yield (8.9 vs. 2.1; *p* < 0.001) and lymph node metastases (25.3% vs. 4.7%; *p* < 0.001) were significantly higher in the pCLND group. In contrast, the lymph node ratio was lower in the pCLND group (0.3 vs. 0.6; *p* < 0.001).

Patients were classified according to the ATA stratification for risk of structural disease recurrence. Patients in the pCLND group were ranked more frequently in the intermediate class of risk (43% vs. 9.4%; *p* < 0.001) than those in the other group, whereas low and high risk classes were similar between the two groups.

Radioactive iodine (RAI) therapy was administrated significantly more often in the pCLND group (92.4% vs. 80.6%; *p* = 0.019).

Overall, 22 (5.5%) patients experienced recurrent disease; 16 of these were localized in the central compartment, whereas the others were in the lateral neck compartment. The crude incidence of disease recurrence was similar between the two groups: 5.6% in The TT group and 5.1% in the pCLND group (*p* = 0.936). Log-rank test on Kaplan–Meier curves, reported in Figure 2a, did not show any significant difference between the two groups (*p* = 0.0883; HR 0.9267, 95% CI 0.3203–2.6814).

A subset analysis was performed considering the ATA stratification for risk of disease recurrence, as reported in Table 3. Considering the patients in the intermediate and high class of risk, the incidence of recurrent disease was significantly lower in the pCLND group (5.4% vs. 21.2%; *p* = 0.0392) than in the TT group. On the other hand, no significant differences were observed in the low class of risk. Log-rank test on Kaplan–Meier curves for patients in intermediate and high class of risk, reported in Figure 2b, demonstrated a significant difference between the two curves (*p* = 0.0439; HR 0.3299, 95% CI 0.1092–0.9967).

Overall, occult lymph node metastases in our series were found in 35 (8.8%) patients. As reported in Table 4, the incidence of disease recurrence was 20% in patients with lymph node metastases (pN+), 2.8% in patients with uninvolved lymph nodes in which an evaluation of N status was possible because at least one lymph node was excised (pN0), and 5% in patients in which N status was not assessed (pNx) (*p* < 0.001). The log-rank test on Kaplan–Meier curves representing patients with and without lymph node metastases (Figure 3) demonstrated a significant difference between the two groups (*p* < 0.001; HR 15.160, 95% CI 3.444–566.7289).

## 4. Discussion

In this work, we focused our attention on prophylactic CLND and the influence of occult lymph node metastases on the prognosis in patients with DTC. 

The first issue to consider is that in our study, there was an intrinsic bias in the selection of patient candidates for pCLND: in fact, this treatment was reserved to cases in which preoperative evaluation including family history, physical examination, US features, FNAC findings, and intraoperative examination suggested a tumor with potentially aggressive behavior.

In our series, the patients who underwent pCLND more frequently had a cytologic diagnosis of PTC and, in addition, were younger than the patients who underwent TT alone. These aspects can be explained by the fact that the surgeon chooses to perform a pCLND in the case of certainty of a malignant nodule: in this case, the operator feels authorized to perform an aggressive intervention to achieve a radical excision of the tumor, mostly in younger patients with a longer life expectancy. On the other hand, when facing patients in whom preoperative diagnosis is uncertain, the surgeon seems inclined to more conservative surgery, preferring a prudent approach to prevent postoperative complications. 

However, unlike what could be expected considering the inherent bias of our study in the selection of the surgical approach, at histopathological examination, the two groups appeared comparable: the nodule size and the presence of aggressive features including multicentricity, angioinvasivity, and extrathyroidal extension were similar between the two groups. 

At this point, a consideration should be made with regard to the indication for pCLND. The ATA guidelines suggest this approach in cases of advanced primary tumors (T3 or T4). Even if our indications were larger than those purposed by the ATA, our work seems to indicate that preoperative and intraoperative evaluation have low reliability in establishing what tumors have pathologically aggressive features that could benefit from a pCLND.

The incidence of tall cell carcinoma, which has been largely described as an aggressive variant of PTC [29,30,31], was significantly higher in the pCLND group. This fact could be explained by the higher prevalence of the FNAC diagnostic for malignancy in the pCLND group: in fact, tall cell carcinoma is associated with considerable alterations of the cells that result in a higher incidence of Tir4 and Tir5. In contrast, the higher incidence of follicular variant of PTC in the TT group could be explained by the fact that this subtype of tumor presents less cellular abnormalities, thus is more often associated with inconclusive or negative FNAC, and consequently with a more conservative approach.

As expected, the incidence of lymph node metastases was higher in the pCLND group (25.3%) than in the TT group (4.7%). However, if we consider only the 99 patients in the TT group in which an evaluation of the N status was possible because at least one lymph node was excised, the real incidence of lymph node metastases was 15.2%, thus similar to the other group. Furthermore, this finding explains the fact that the lymph node ratio was significantly higher in the TT group, where the denominator of the fraction was smaller because a smaller number of lymph nodes was excised. These findings suggest that the incidence of lymph node metastases is higher when a larger number of lymph nodes is excised [24,28].

Considering the ATA risk stratification for structural disease recurrence, patients in the pCLND group were more often classified in the intermediate risk group. This could be explained by the fact that pCLND allows for more accurate staging of the tumor, ensuring a better assessment of the N status; in fact, the intermediate class of risk includes tumors in which more than five lymph nodes are involved. Thus, it is likely that pCLND allows to upstage tumors that otherwise would have been classified as low risk tumors, as already reported in the literature [28,32,33]. This fact also explains the higher incidence of patients who underwent RAI therapy after surgery in the pCNLD group.

Overall, our work failed to demonstrate an advantage on prognosis in patients who underwent pCLND. However, if we consider only patients at intermediate and high risk of recurrence, pCLND significantly improved the disease-free survival. We think that this is the key point because tumors at low risk of recurrence have a good prognosis, thus pCLND could be considered an overtreatment that does not modify the course of the disease; on the other hand, the real value of pCLND is expressed in tumors at intermediate and high risk of recurrence, which benefit from an aggressive surgery, with a reduction of recurrence rate. 

These findings are in accordance with a recent meta-analysis by Zhao et al., which included 22 studies with over 6000 patients, where pCLND proved to be effective in reducing the risk of loco-regional recurrence [34]; another meta-analysis regarding pCNLD in patients who underwent hemithyroidectomy was consistent with this result [35].

The secondary outcome of our work was to assess whether pCLND was burdened by a higher incidence of postoperative complications. In our series, the occurrence of hypoparathyroidism and RLN injury was higher in the pCLND group, but this difference was not significant, suggesting that pCLND could be a safe procedure with an acceptable incidence of complications. However, this finding should be carefully considered and contextualized: in fact, the same meta-analysis of Zhao et al. that we previously reported, demonstrated a higher incidence of transient and permanent hypoparathyroidism and of transient RLN injury [34].

Furthermore, we must underline that the overall incidence of permanent hypoparathyroidism (9.8%) in our study was higher than the ones usually reported in the literature. As already stated in the Methods section, we defined hypoparathyroidism on the basis of PTH value; probably, this assessment overestimates the incidence of hypoparathyroidism compared to the centers that use only serum calcemia as a criterion, which can be easily influenced from oral calcium supplementation. 

Finally, we considered the influence of lymph node metastases on prognosis. When excluding the patients in which the N status was not assessed because no lymph node was excised (pNx), the incidence of recurrent disease was considerably higher in patients with lymph node metastases (pN+), reaching up to 20% than in patients with uninvolved lymph nodes (pN0), with an incidence of 2.8%. It is also interesting to observe that in the group in which the N status was not assessed (pNx), the incidence of recurrences was almost twice (5%) that in patients in the pN0 group, perhaps suggesting that some of these recurrences could have been avoided if a lymphectomy had been performed.

As already mentioned, current guidelines are discordant regarding pCLND. In the ATA guidelines published in 2015, pCLND assumes a marginal role in the treatment of differentiated thyroid carcinoma [1]. The latest NCCN guidelines published in 2019 have eliminated, compared with the previous edition, pCLND in DTC [27]. Considering the negative impact on the prognosis and the high incidence of occult lymph node metastases, and taking into account the difficulties in establishing preoperatively and intraoperatively what tumors could have aggressive behavior, we think that indications for pCLND could be revised in order to achieve more efficacious treatment of aggressive tumors. Such considerations are in accordance with a recent meta-analysis of Zhao et al., which included over 4000 patients, reporting a poor sensitivity of US in detecting metastases of the central compartment (pooled sensitivity of 33%, range 10–57%) with an incidence of lymph node metastases of 48%, suggesting for these reasons that indications to pCLND should be extended to all patients with DTC [36].

This study has some limitations. First, this is a single center, retrospective study. The study was performed in an endemic iodine deficient region, with a high incidence of autoimmune thyroiditis; therefore, the generalization of our results to other populations should be made carefully. Finally, the real incidence of disease recurrence could be underestimated in our study, considering that the mean follow-up is 55.4 months, and that these kind of tumors are generally indolent, and recurrences can appear up to 10 years after surgery.

## 5. Conclusions

The selection of patients for prophylactic CLND is problematic due to the low accuracy of preoperative and intraoperative evaluation in establishing what tumors are aggressive and could benefit from aggressive surgery. Prophylactic CLND is a safe procedure, with an acceptable incidence of complications, comparable to that of patients who undergo thyroidectomy alone. Our study demonstrated that pCNLD allows for more accurate staging of the tumor and reduces the incidence of recurrent disease in patients with intermediate and high risk DTC. We think that indications for pCNLD should be revised by the main guidelines, in consideration with the latest evidence in the literature.

## Figures and Tables

**Figure 1 cancers-12-01658-f001:**
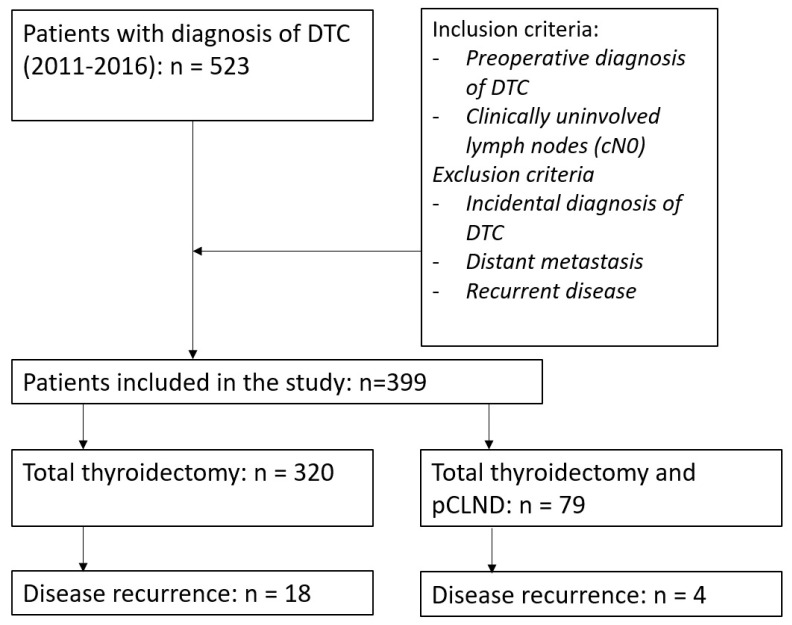
Design of the study. DTC: differentiated thyroid carcinoma; pCLND: prophylactic central lymph node dissection.

**Figure 2 cancers-12-01658-f002:**
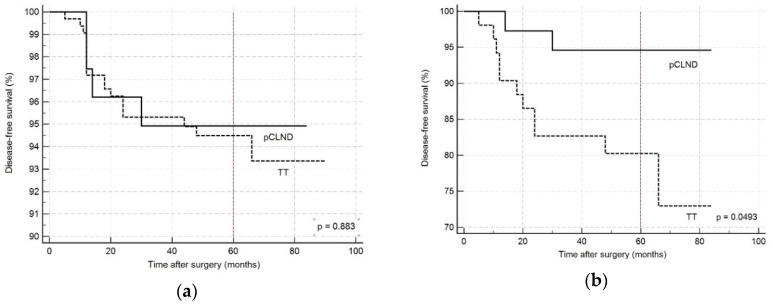
Kaplan–Meier curves estimating disease-free survival according to surgical procedure, (**a**) including all the patients regardless of ATA class risk of recurrence: *p* = 0.883, HR 0.9267 (95% CI 0.3203–2.6814); (**b**) including only patients with moderate and high risk for disease recurrence according to ATA guidelines: *p* = 0.0493, HR 0.3299 (95% CI 0.1092–0.9967). pCLND: prophylactic central lymph node dissection; TT: total thyroidectomy; HR: hazard risk.

**Figure 3 cancers-12-01658-f003:**
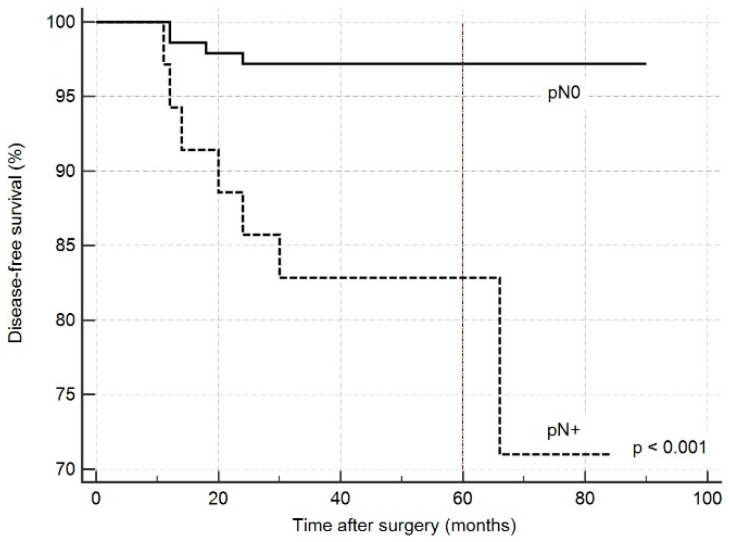
Kaplan-Meier curves estimating overall disease-free survival according to N status: *p* < 0.001 (HR 15.160, 95% CI 3.4445–66.7289). pN0: No evidence of lymph node metastases at pathological examination. pN+: Lymph node metastases at pathological examination.

**Table 1 cancers-12-01658-t001:** Univariate analysis of demographic, preoperative, operative data, and outcomes of patients with differentiated thyroid carcinoma and clinically uninvolved lymph nodes.

	Patients (*n* = 399)	TT Group (*n* = 320)	pCLND Group (*n* = 79)	*p*
Sex				0.1493
Male	101 (25.3%)	86 (26.9%)	15 (19.0%)	
Female	298 (74.7%)	234 (73.1%)	64 (81.0%)	
Age, years	50.5 ± 14.4 (15–83)	52.4 + 14.0	42.6 ± 13.5	*p* < 0.001
Hyperthyroidism	28 (7%)	25 (7.8%)	3 (3.8%)	0.314
Autoimmune thyroiditis	146 (36.6%)	97 (30.3%)	49 (62.0%)	*p* < 0.001
US findings				
Multinodular disease	250 (62.7%)	204 (63.8%)	46 (58.2%)	0.436
Peri- and intra-vascularization of the nodule	243 (60.9%)	195 (60.9%)	48 (60.8%)	0.920
Hypoechoic nodule	70 (17.6%)	48 (15%)	22 (27.8%)	0.011
Microcalcification	34 (8.5%)	29 (9.1%)	5 (6.3%)	0.579
FNAC				<0.001
Tir1-2	73 (18.3%)	66 (20.6%)	7 (8.9%)	
Tir3	148 (37.1%)	131 (40.9%)	17 (21.5%)	
Tir4-5	178 (44.6%)	123 (38.4%)	55 (69.6%)	
Operative time, minutes	94.4 ± 22.2	92.5 ± 22.6	102.3 ± 18.9	*p* < 0.001
Postoperative stay, days	2.8 ± 1.1	2.8 ± 1.0	3.0 ± 1.3	0.068
Transient hypoparathyroidism	136 (34.1%)	102 (31.9%)	34 (43.0%)	0.081
Permanent hypoparathyroidism	39 (9.8%)	27 (8.4%)	12 (15.2%)	0.109
Transient RLN injury	10 (2.5%)	7 (2.2%)	3 (3.8%)	0.675
Permanent RLN injury	3 (0.8%)	2 (0.6%)	1 (1.3%)	0.891
Postoperative bleeding	5 (1.3%)	4 (1.3%)	1 (1.3%)	0.580
RAI therapy	331 (82.9%)	258 (80.6%)	73 (92.4%)	0.019
Follow-up, months	55.4 ± 15.9	56.1 ± 16.1	52.6 ± 15.2	0.087
Recurrent disease	22 (5.5%)	18 (5.6%)	4 (5.1%)	0.936

TT: total thyroidectomy; pCLND: prophylactic central compartment lymph node dissection; US: ultrasound; FNAC: fine-needle aspiration cytology; RLN: recurrent laryngeal nerve; RAI: radioactive iodine. Continuous variables are reported as the mean ± standard deviation of the mean.

**Table 2 cancers-12-01658-t002:** Univariate analysis of pathological data of patients with differentiated thyroid carcinoma and clinically uninvolved lymph nodes.

	Patients (*n* = 399)	TT Group (*n* = 320)	pCLND Group (*n* = 79)	*p*
Nodule size (mm)	16.8 ± 10.9	16.9 ± 11.1	16.3 ± 9.8	0.681
Thyroid weight (gr)	26.7 ± 22.5	27.4 ± 24.2	24.1 ± 13.3	0.252
Histotype				<0.0001
PTC	151	115 (35.9%)	36 (45.6%)	
FV-PTC	133	120 (37.5%)	13 (16.5%)	
TCV-PTC	40	20 (6.3%)	20 (25.3%)	
FTC	53	47 (14.7%)	6 (7.6%)	
HCC	21	18 (5.6%)	3 (3.8%)	
Low differentiated carcinoma	1	0	1 (1.3%)	
Microcarcinoma	109 (27.3%)	87 (27.2%)	22 (27.8%)	0.981
Multicentricity	151 (37.8%)	117 (36.6%)	34 (43%)	0.350
Angioinvasivity	15 (3.8%)	13 (4.1%)	2 (2.5%)	0.756
Extrathyroidal extension	27 (6.8%)	23 (7.2%)	4 (5.1%)	0.672
LN yield	5.1 ± 5.3	2.1 ± 1.1	8.9 ± 6.0	*p* < 0.001
LN metastasis	35 (8.8%)	15 (4.7%)	20 (25.3%)	*p* < 0.001
Number of positive LN	0.5 ± 1.3	0.2 ± 0.7	0.8 ± 1.7	0.006
LN ratio	0.5 ± 0.3	0.6 ± 0.2	0.3 ± 0.3	0.004
ATA risk class of disease recurrence				*p* < 0.001
Low	310 (77.7%)	268 (83.8%)	42 (53.2%)	
Medium	64 (16%)	30 (9.4%)	34 (43.0%)	
High	25 (6.3%)	22 (6.9%)	3 (3.8%)	

TT: total thyroidectomy; pCLND: prophylactic central compartment lymph node dissection; PTC: papillary thyroid carcinoma; FV-PTC: follicular variant of PTC; TCV-PTC: tall cell variant of PTC; FTC: follicular carcinoma; HCC: hurtle cell carcinoma; LN: lymph node; ATA: American Thyroid Association. Continuous variables are reported as the mean ± standard deviation of the mean.

**Table 3 cancers-12-01658-t003:** Univariate analysis of recurrent disease in patients stratified for risk of structural recurrence according to the American Thyroid Association guidelines.

ATA Risk	Low			Medium–High		
	TT	pCLND	Total	TT	pCLND	Total
Patients	268	42	310	52	37	89
Recurrent disease	7 (2.6%)	2 (4.8%)	9 (2.9%)	11 (21.2%)	2 (5.4%)	13 (14.6%)
*p* (TT vs. CLND)	0.441			0.0392		

ATA: American Thyroid Association guidelines; TT: total thyroidectomy; pCLND: prophylactic central compartment lymph node dissection.

**Table 4 cancers-12-01658-t004:** Univariate analysis of recurrent disease in patients stratified according to N status.

	pNx	pN0	pN+
Patients	221	143	35
Disease recurrence	11 (5%)	4 (2.8%)	7 (20%)

pNx: pathological nodal status not assessed; pN0: lymph nodes uninvolved at pathological examination; pN+: lymph node metastases at pathological examination. Statistical significance: overall *p* < 0.001; pNx vs. pN0: *p* = 0.1132; pNx vs. pN+: *p* < 0.001; pN0 vs. pN+ *p* < 0.001.

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
