# Peer review of "Prophylactic Central Lymph Node Dissection Improves Disease-Free Survival in Patients with Intermediate and High Risk Differentiated Thyroid Carcinoma: A Retrospective Analysis on 399 Patients"

_cancers, 2020, doi:10.3390/cancers12061658_

Round 1

Reviewer 1 Report

The aim of this retrospective study was to analyze the prognostic value of prophylactic central node dissection (pCNLD) in differentiated thyroid cancer on risk of recurrent disease, the influence of occult node metastases and the risk of postoperative complications. A cohort of 399 patients with differentiated thyroid carcinoma (DTC) and clinically uninvolved lymph nodes, treated between January 2011 and December 2016 was analyzed. The authors concluded that available methods for pre- and perioperative selection of patients who could benefit from central node clearance due to aggressive features of tumor have low accuracy and that central node clearance is a safe procedure that allows accurate staging of the tumor and reduces the incidence of recurrence in patients with intermediate and high-risk DTC. The authors also proposed a revision of current guidelines concerning indications to perform pCNLD.

The results are interesting but I have some comments and concerns.

  1. The main weakness of this Ms. is the selection of patients. To evaluate the generalizability of the results, the description of the intrinsic bias must be expanded. The authors mention several factors such as family history and ultrasound findings.
    1. Heredity is unusual in DTC – was there any familial accumulation?
    2. Is it true that no incidental findings of thyroid cancer were included in this study? The percentage of hyperthyroidism was high – wasn´t it the main indication in some of the cases? Also, the percentage of microcarcinoma was 27 and 28%, please comment on this.
    3. Very few cases were verified by cytology preoperatively, how many had had fine needle biopsy?
    4. The reporting system for cytology, TIR 1-5, needs explanation
    5. Was ultrasound performed only by one experienced surgeon?
    6. How many surgeons were responsible for the surgical procedures? Was the choice of pCNLD or not dependent on personal preferences?
    7. The number of N+ is dependent on the comprehensiveness of the lymph node sectioning (Ref. Haglund et al, Endocrine pathology 2016). In this study, patients were defined as pN0 if at least one lymph gland was excised. Please add information about the local routine, number of pathologists involved, and the classification of micrometastases.
  2. The risk of permanent hypoparathyroidism was high, 15.2% and the conclusion about pCNLD as a safe procedure has to be questioned. However, several definitions for hypoparathyroidism exists, I suggest you should add your definition of permanent hypoparathyroidism and also address this issue in the discussion section.
  3. The description of the follow-up needs to be expanded
    1. time intervals?
    2. both ultrasound and biochemistry?
    3. thyrogen stimulated thyroglobulin measurements?

Reviewer 2 Report

This paper looked at the role of prophylactic central lymph node dissection (pCLND) in patients with thyroid carcinoma, and the authors found that pCLND improved disease-free survival in patients with intermediate and high risk differentiated thyroid carcinoma.

  1. In Abstract, line 14, a should be placed before matter.
  2. In Abstract, line 16, who should replace that.
  3. In Abstract, line 17 and throughout the paper, metastases should replace metastasis.
  4. In Abstract, lines 21 and 22, the number of patients adds up to 419, not 399. In the Results and Table 1, 399 patients are noted.
  5. In Introduction, line 34, a bad course is unclear.
  6. In Introduction, line 38 and throughout the paper, metastasis should be metastases and extra thyroidal should be extrathyroidal.
  7. In Introduction, line 42, purposed should be proposed.  In that line and continuing to lines 43 and 44 on page 2, I suggest: grouped as low, intermediate...requiring careful follow-up and aggressive treatment in cases of.
  8. In Introduction, lines 45 and 46, I suggest: One of the outstanding issues is the role of...in cases of clinically uninvolved lymph nodes.  The incidence.
  9. In Introduction, line 51, purposed should be changed to proposed.
  10. In Introduction, line 55, and throughout the paper, metastasis should be metastases
  11. In Introduction, line 62, I suggest: occult lymph node metastases on the prognosis.
  12. In Methods, lines 63 and 71, were should be changed to we.
  13. In Methods, line 67, that should be changed to who.
  14. In Methods, line 72, I suggest: in the second group, patients underwent thyroidectomy.
  15. In Methods, line 77, I suggest adding lymph before node.
  16. In Methods, line 79, I suggest: included clinical history, physical examination, and blood tests
  17. In Methods, line 81 and 82, the should be placed before results and the abbreviations (AIT, AME, SIE, and SIAPEC-IAI) should be defined.
  18. In Methods, lines 91 and 92, I suggest: of a highly suspicious nodule based on family history, physical examination, and the US features of the nodule.
  19. In Methods, line 93, I suggest: The surgical procedures were extracapsular total thyroidectomies. The RLN were.
  20. In Methods, lines 98 and 100, case should be changed to cases.
  21. In Methods, lines 101 and 102, I suggest: consisted of excision of all...included in Level VI and Level VII.
  22. In Methods, line 103,  I suggest: neck. The anatomical limits of the dissection were represented.
  23. In Methods, lines 106 and 107, I suggest: fascia. The central compartment included the pretracheal...paratracheal, and paralaryngeal....Level VII was comprised of the.
  24. In Methods, lines 112 and 113, Haematoxylin-Eosin should be haematoxylin-eosin.
  25. In Methods, line 115, case should be cases.
  26. In Methods, line 117 and throughout the paper, Extra thyroidal should be Extrathryoidal.
  27. In Methods, line 118, I suggest: examination. Vascular invasion was defined as invasion of.
  28. In Methods, line 121, I suggest adding the before lymph.
  29. In Methods, line 130, I suggest adding an before endocrinologist.
  30. In Methods, line 132, the sentence starting with Postoperative is not clear.
  31. In Methods, lines 136 and 137, I suggest: retrospectively, including patients who underwent.
  32. In Methods, line 141, a should be placed before p-value.
  33. In Results, line 149, remnant should be changed to remaining.
  34. In Results, line 150 and throughout the paper, familial should be replaced by family.
  35. In Results, line 152, I suggest: The surgical procedure consisted of total thyroidectomy alone...patients and total.
  36. In Results, lines 154 and 155, the should be placed before surgical. In the next sentence, I suggest:  Those who underwent prophylactic.
  37. In Results, line 156, from should be changed to with.
  38. In Results, line 158 and throughout the paper, the should be added before pCLND.
  39. In Results, lines 158 and 159, I believe that the operative times and the postoperative stays were averages.  This should be clarified in the text and in Table 1 (where I believe that averages were reported +/- one standard deviation). 
  40. In Table 1, in Peri- and intra-vascularization of the nodule, first column on the left, 243/399 = 60.9%, not 81.3%.
  41. In Results, line 170, the should be placed before pCLND and tall.
  42. In Results, line 171,  I suggest: nearly quadruple that in the...incidence of the follicular.
  43. In Results, line 172,  a can be cut. Following 37.5%), I suggest adding: when compared to the TT group.
  44. In Results, line 176, I suggest adding lymph nodes after VII and adding a comma after group.
  45. In Results, line 186, following p = 0.019) I suggest adding: than in the TT group.
  46. In Table 2, in Angioinvasivity, second column, 13/ 320 =4.1%.
  47. In Results, line 193, a can be cut.
  48. In Results, line 196, didn't should be changed to did not.
  49. In Discussion, line 231, the comma after CLND can be cut.
  50. In Discussion, line 236, potential should be changed to potentially.
  51. In Discussion, line 237, I suggest: series, the patients...pCLND more frequently had a cytologic.
  52. In Discussion, the sentence starting in line 238 with These aspects is a run-on sentence.
  53. In Discussion, line 242, I suggest changing By the other side to On the other hand.
  54. In Discussion, line 249, I suggest: indications for pCLND.
  55. In Discussion, line 250, case should be changed to cases.
  56. In Discussion, line 253, pathological should be changed to pathologically.
  57. In Discussion, line 254, Instead can be cut.
  58. In Discussion, line 260, to should be changed to with.
  59. In Discussion, line 263 and throughout the paper, the should be added before TT.
  60. In Discussion, line 267, I suggest: a smaller number of lymph nodes was excised.
  61. In Discussion, lines 268 and 269, the sentence starting with These findings is not clear.
  62. In Discussion, line 270, the should be added before ATA.
  63. In Discussion, line 272, I suggest: pCLND allows for more.
  64. In Discussion, line 277, we should change on to regarding.
  65. In Discussion, line 281, by the other side should be changed to on the other hand.
  66. In Discussion, lines 284 and 286, metanalysis should be meta-analysis.
  67. In Discussion, line 284 and 292, I suggest: meta-analysis by Zhao et al.
  68. In Discussion, line 290, RLNs should be RLN.
  69. In Discussion, line 300,  I suggest (5%) that in patients in the pN0 group.
  70. In Discussion, line 303, this approach is not clear.
  71. In Discussion, lines 304 and 305, I suggest: 2019, have eliminated pCLND in DTC.
  72. In Discussion, line 308, to should be changed to for. At the end of that sentence, I suggest: revised in order to achieve more efficacious treatment of aggressive tumors.
  73. In Discussion, lines 310 and 311, I suggest: this is a single center, retrospective study.  The study was performed in an endemic.
  74. In Discussion, line 312, I suggest: results to other populations should.
  75. In Conclusions, line 314, I suggest: patients for prophylactic.
  76. In Conclusions, line 318, a should be changed to for.
  77. In Conclusion, lines 319 and 320, I suggest: indications for pCLND should be revised in consideration with the latest.
  78. The references should be written according to Cancers style.

Reviewer 3 Report

In this work, Medas et al. evaluated the impact of prophylactic central lymph node dissection in a restrospective cohort of patients affected by differentiated thyroid carcinoma, in order to assess the effectivness of current guidelines. In particular, They evaluated three main endpoints: (1) incidence of recurrence after pCLND; (2) post-operative complications after pCLND and (3) prognostic value of occult node metastasis.

In conclusion, the Authors stated that pCLND improves tumor staging and reduces the risk of recurrence in patients with intermidiate and high risk of recurrence, suggesting a revision of current guidelines.

The conclusion of this work are supported by recently published papers such as Zhao et al, Eur J Radiology 2019 that I suggest to include in the discussion section. However the study presents several concerns, especially in manuscript structure and form and at some istances also in content.

-Introduction section should be revised. First of all the incidence of local recurrence should be better described. The Authors reported a wide range of recurrence (2 – 30%) without explaining the conditions influencing this distribution and also the reported references are not updated.

Then, to define patients subdivision in different groups They simply refer to "pathological findings", please better specified the main parameters taken into account. By reading the introduction it seems that Authors follow ATA guidelines to assign patients to the different groups but this is not clear passing to methods section.

-Methods should be revised too, the paragraph is very confused in its current form, informations are mixed and difficult to follow. I suggest to subdivide this section into different paragraphs in order to clarify the study design and the parameters considered, both in inclusion criteria for study cohort determination and in the ones usefull to determine the inclusion in TT or pCLND groups. Patients are grouped by applying retrospectively the ATA criteria published in 2015? Since it is difficult to understand the study design, it is difficult also to evaluate the strengthen of the results. A schematic rapresentation of study design would be very useful.

-the time indicated for follow up of 4 years is not so long considering this kind of lesions that are generally indolent and especially for evaluation of recurrence or appearance of metastases several studies reach 10 years of FUP in order to obtain more reliable results.

- In methods section, the Authors declared that all the 399 patients included in the study have clinically uninvolved lymph nodes(cN0), then in results table 4 They subdivided the cohort in pNx, pN0, pN+ on the basis of pathological diagnosis. These data seems to be inconsistent.

Minor Concerns:

-in the Abstract please correct the number of patients included in the pCLND group.

-a wide editing of language is required.

Round 2

Reviewer 1 Report

nothing to add

Reviewer 3 Report

The Authors replied to all the concerns indicated in the previous review report. This work still have some bias, that as described by the Authors in the discussion, may limit the generalizability of the results in particular in relation to the main outcome. However the manuscript is now easier to follow.

Still remain several typing errors, please correct them.